# Synthesis of Ammonium-Based ILs with Different Lengths of Aliphatic Chains and Organic Halogen-Free Anions as Corrosion Inhibitors of API X52 Steel

**DOI:** 10.3390/ijms24087613

**Published:** 2023-04-20

**Authors:** Emiliano Cornejo Robles, Octavio Olivares-Xometl, Natalya V. Likhanova, Paulina Arellanes-Lozada, Irina V. Lijanova, Víctor Díaz-Jiménez

**Affiliations:** 1CIITEC, Instituto Politécnico Nacional, Cerrada Cecati S/N, Colonia Santa Catarina de Azcapotzalco, Ciudad de Mexico 02250, Mexico; emilianorobles043@gmail.com; 2Facultad de Ingeniería Química, Benemérita Universidad Autónoma de Puebla, Av. San Claudio y 18 Sur, Ciudad Universitaria, Col. San Manuel, Puebla 72570, Mexico; oxoctavio@yahoo.com.mx (O.O.-X.);; 3Programa de Investigación y Posgrado, Instituto Mexicano del Petróleo, Eje Central Norte Lázaro Cárdenas No. 152, Col. San Bartolo Atepehuacan, G. A. Madero, Ciudad de Mexico 07730, Mexico

**Keywords:** ionic liquids, API X52, corrosion inhibitors, hydrochloric acid, inhibition mechanism

## Abstract

In the present work, synthesis and characterization of 15 ionic liquids (ILs) derived from quaternary ammonium and carboxylates were carried out in order to proceed to their evaluation as corrosion inhibitors (CIs) of API X52 steel in 0.5 M HCl. Potentiodynamic tests confirmed the inhibition efficiency (*IE*) as a function of the chemical configuration of the anion and cation. It was observed that the presence of two carboxylic groups in long linear aliphatic chains reduced the *IE*, whereas in shorter chains it was increased. Tafel-polarization results revealed the ILs as mixed-type CIs and that the *IE* was directly proportional to the CI concentration. The compounds with the best *IE* were 2-amine-benzoate of N,N,N-trimethyl-hexadecan-1-ammonium ([THDA^+^][^−^AA]), 3-carboxybut-3-enoate of N,N,N-trimethyl-hexadecan-1-ammonium ([THDA^+^][^−^AI]), and dodecanoate of N,N,N-trimethyl-hexadecan-1-ammonium ([THDA^+^][^−^AD]) within the 56–84% interval. Furthermore, it was found that the ILs obeyed the Langmuir adsorption isotherm model and inhibited the corrosion of steel through a physicochemical process. Finally, the surface analysis by scanning electron microscopy (SEM) confirmed less steel damage in the presence of CI due to the inhibitor–metal interaction.

## 1. Introduction

Not only due to its physicochemical properties but also to economic factors and easy access, carbon steel is widely used in different sectors of the oil-and-gas industry. Despite displaying good mechanical resistance in the case of different shear-stress sources, this material is susceptible to suffering corrosion damage when exposed to corrosive media. API X52 is a type of carbon steel that is employed in the production of pipelines for transporting oil and gas [1]. At the industrial level, there are different aggressive media for API X52 steel, which have been classified as sweet, sour, and acid, where organic and inorganic substances, salts, and gases, among other compounds, can be found [2,3,4]. In this context, acid media are employed to increase the amount of extracted oil; this technique is known as stimulation through acidification, where acid solutions ranging from 5 to 28 wt.% of HCl are injected in order to modify the permeability of the reservoir rock, thus easing the oil flow [5]. Notwithstanding, the use of such a technique provokes a significant problem of internal corrosion in oil pipelines due to the presence of HCl [6]. In acid media, the medium aggressiveness depends on variables such as pH, temperature, flow regime, steel composition, pressure, etc. [7,8]. For this reason, steel corrosion is a complex process that is hard to control and understand [9,10]. 

Because of the importance of extending the useful life of steel, different methods against corrosion have been developed [11]. In this context, the synthesis and application of organic-type corrosion inhibitors (CIs) in different corrosive media is very common due to their easy use and low cost [12,13,14]. Notwithstanding, their employment is reduced to polar electrolytes because of their limited solubility, high volatility, non-biodegradability, and environment-hazardous features [15]. These disadvantages and present environmental regulations have promoted the implementation of new Cis known as ionic liquids (Ils), which have earned a well-deserved place due to their versatile structures, where heteroatoms such as nitrogen and oxygen define electronic densities and, at the same time, either aliphatic chains or aromatic rings provide hydrophobic features. Within the large number of possible combinations, halide-free Ils are specially promising as environmentally friendly Cis due to their green properties [16]. Among the different Ils, imidazolium-based compounds have been employed to mitigate the corrosion effects on metallic surfaces in different processes related to oil, desalination, and acid-cleaning applications [17,18]. In most cases, halogen-containing anions form part of ILs, and only a few works have reported on the use of halogen-free ILs, like the one by Chen et al., where compounds with ammonium-dibutyl-dithiophosphate anions displayed anti-corrosion performance [19,20]. Recently, ammonium-based ILs have reached the vanguard position thanks to their excellent inhibition properties, chemical stability, biodegradability, and low-cost production [21,22,23,24]. In general, tertiary amines are used to synthesize ammonium-based ILs, where the length of cationic aliphatic chains plays a major role in the inhibition process, without discarding the influence of the anionic part. Heteroatoms and π electrons of dimeric quaternary ammonium salts have been identified as favorable factors for efficient corrosion protection in strong acid media [25]. For instance, Likhanova et al. showed that the molecular orientation to metallic surfaces through high-density zones, represented either by organic (adipate) or inorganic (ethyl-sulfate) anionic structures of ammonium-based ILs, enhanced the anticorrosion effect (70–80%) [26]. In this context, another research work compared the aliphatic chains [22,27] of anionic parts formed by different dicarboxylic acids and found Gibbs-adsorption energy values equal to 37.2, 37.2, and 35.1 kJ mol^−1^, which were attributed to mixed adsorption (physicochemical) [21]. As for triethyl-methyl-ammonium ILs with different anions (either with long aliphatic chains or aromatic rings), they displayed anticorrosion activity in acid medium (H_2_SO_4_) above 70% [28]. At this point, it is worth emphasizing that ammonium-based ILs with organic anions are in the trend of green-chemistry conceptuality to diminish the corrosion effect [29]. Zhu et al. synthesized tetrabutyl-ammonium ILs with 14 different amino-acid anions and evaluated their corrosion resistance, finding that the ILs exhibited remarkable friction reduction and anti-wear features [30]. In this context, Aslam et al. reported that amino-acid-ester-salt/saccharine-based ILs worked as powerful green Cis of mild steel in acidic media [31]. From the different types of Ils that have been employed as Cis, most contain heteroatoms (phosphorus, nitrogen, oxygen, and nitrogen), functional groups (–C=N–, –NH_2_, –OH, –OCH_3_, –SH, etc.), and alkyl chains with different lengths. As a whole, this IL chemical configuration will define the hydrophilic and nucleophilic parts and establish the interface energy barrier due to either physical or chemical adsorption [32]. Based on the aforementioned, the search for new CIs for the industry is fundamental in order to extend the useful life of steel alloys employed in the transport and refining of oil. As part of these efforts, the present work deals with the synthesis, characterization, and evaluation of a series of 15 new ammonium-derived ILs as CIs of API X52 steel in 0.5 M HCl. The evaluated ILs are different from conventional ones because they present carboxylates and not halides in their anionic structure. The present study emphasized the importance of the chemical configuration of the cation and anion in the ILs, which consisted of different alkyl-chain lengths and functional groups such as amine, ammonium, benzoate, and carboxyl. A corrosion-inhibition mechanism supported by techniques such as polarization resistance (Rp), potentiodynamic polarization (PDP), electrochemical impedance spectroscopy (EIS), scanning electron microscopy/energy dispersive spectroscopy (SEM/EDS), and DFT theoretical calculations (B3LYP/6-311G) was proposed. 

## 2. Results and Discussion

### 2.1. Characterization of ILs

Table 1 shows the ILs that were synthesized to be evaluated as CIs.

Prior to the evaluation as CIs, the ILs were characterized by the proton (^1^H) and carbon (^13^C) nuclear magnetic resonance (NMR) technique by employing a piece of JEOL Eclipse-300 equipment, which uses trimethylsilane (TMS) as a standard for the chemical displacements (ppm) and deuterated chloroform as solvent at ambient temperature. The synthesis of the ILs was confirmed by the FT-IR technique by means of a Nicolet Nexus 470 FT-IR spectrophotometer in attenuated-total-reflection (ATR) mode. The signals obtained for the studied compounds were the following:

Methyl-carbonate of N,N,N-trimethyl-hexadecan-1-ammonium ([THDA^+^][^−^MC]; C_21_H_45_NO_3_; pasty brown dough; 98.7% yield): IR (KBr, cm^−1^): 3427, 2920, 2851, 2797, 2663, 1665, 1472, 961, 721. ^1^H NMR (CDCl_3_) δH (ppm): 0.88 (t, 3H, CH_3_), 1.25 (m, 26H, CH_2_), 1.80 (d, 2H, CH_2_), 2.67 (s, 2H, CH_2_), 3.22 (s, 3H, CH_3_-O), 4.49 (s, 9H, CH_3_-N). ^13^C NMR (CDCl_3_) δH (ppm): 14.04 (CH_3_), 22.61 (CH_2_), 23.13 (CH_2_), 26.75 (CH_2_), 29.63 (CH_2_), 31.85 (CH_2_), 53.33 (O-CH_3_), 58.10 (CH_3_-N), 66.83 (CH_2_-N), 162.59 (C=O). 

Butyrate of N,N,N-trimethyl-hexadecan-1-ammonium ([THDA^+^][^−^AB]; C_23_H_49_NO_2_; brown powder; 51.5% yield): IR (KBr, cm^−1^): 3401, 2929, 1560, 1486, 1417, 966, 717. ^1^H NMR (CDCl_3_) δH (ppm): 0.89 (m, 6H, CH_3_), 1.25 (m, 26H, CH_2_), 1.55 (m, 2H, CH_2_), 1.68 (s, 2H, CH_2_), 2.12 (t, 2H, CH_2_), 3.24 (s, 9H, CH_3_-N), 3.33 (t, 2H, CH_2_-N). ^13^C NMR (CDCl_3_) δH (ppm): 14.42 (CH_2_), 19.91 (CH_2_), 22.82 (CH_2_), 23.27 (CH_2_), 29.85 (CH_2_), 32.06 (CH_2_), 40.04 (CH_2_), 53.30 (CH_3_-N), 66.79 (CH_2_-N), 179.28 (C=O).

3-Carboxybut-3-enoate of N,N,N-trimethyl-hexadecan-1-ammonium ([THDA^+^][^−^AI]; C_24_H_47_NO_4_ brown powder; 51.1% yield): IR (KBr, cm^−1^): 3248, 2918, 1702, 1435, 1216, 915. ^1^H NMR (CDCl_3_) δH (ppm): 0.88 (t, 3H, CH_3_), 1.26 (m, 26H, CH_2_), 1.83 (d, 2H, CH_2_), 2.85 (s, 2H, CH_2_), 3.02 (s, 2H, CH_2_), 3.31 (m, 9H, CH_3_-N), 5.33 (s, 2H, H_2_C=C). ^13^C NMR (CDCl_3_) δH (ppm): 14.09 (CH_3_), 22.66 (CH_2_), 29.34 (CH_2_), 31.90 (CH_2_), 43.05 (CH_2_), 53.37 (CH_3_-N), 66.92 (CH_2_-N), 128.35 (C=C), 134.91 (C=C), 168.48 (COOH), 173.18 (COO^−^). 

3-Carboxy-2,2-dimethylpropanoate of N,N,N-trimethyl-hexadecan-1-ammonium ([THDA^+^][^−^2,2-DSA]; C_25_H_51_NO_4;_ brown powder; 54.3% yield): IR (KBr, cm^−1^): 3100, 2920, 2848, 2675, 1699, 1432, 1311, 1220, 1141, 933, 627. ^1^H NMR (CDCl_3_) δH (ppm): 0.88 (t, 3H, CH_3_), 1.25 (m, 26H, CH_2_), 1.34 (m, 6H, CH_3_), 1.81 (d, 2H, CH_2_) 2.59 (s, 2H, CH_2_-N), 2.85 (s, 2H, CH_2_), 3.25 (s, 9H, CH_3_-N). ^13^C NMR (CDCl_3_) δH (ppm): 14.11 (CH_3_), 22.68 (CH_2_), 24.25 (CH_2_), 26.59 (CH_3_), 29.07 (CH_2_), 29.50 (CH_2_), 31.91 (CH_2_), 40.16 (CH_2_), 44.23 (CH_2_), 53.35 (CH_3_-N), 66.88 (CH_2_-N), 174.72 (COOH), 181.19 (COO^−^). 

2-Amine-benzoate of N,N,N-trimethyl-hexadecan-1-ammonium ([THDA^+^][^−^AA]; C_26_H_48_N_2_O_2_; brown powder; 52.7% yield): IR (KBr, cm^−1^): 3473, 3375, 3196, 2920, 1671, 1484, 1418, 1303, 1243, 917, 751. ^1^H NMR (CDCl_3_) δH (ppm): 0.88 (t, 3H, CH_3_), 1.25 (m, 26H, CH_2_), 1.82 (d, 2H, CH_2_), 3.0 (m, 2H, CH_2_), 3.3 (s, 9H, CH_3_-N), 6.64 (t, 1H, Ar), 6.69 (d, 1H, Ar), 6.81 (s, 1H, Ar), 7.28 (t, 2H, Ar-NH_2_), 7.90 (d,1H, Ar). ^13^C NMR (CDCl_3_) δH (ppm): 14.12 (CH_3_), 22.68 (CH_2_), 24.24 (CH_2_), 26.60 (CH_2_), 29.71 (CH_2_), 31.92 (CH_2_), 53.46 (CH_3_-N), 67.04 (CH_2_-N), 110.07 (C-Ar), 116.35 (C-Ar), 116.83 (C-Ar), 132.03 (C-Ar), 134.78 (C-Ar), 150.92 (C-Ar), 172.61 (C=O). 

Hexanoate of N,N,N-trimethy-lhexadecan-1-ammonium ([THDA^+^][^−^AH]; C_25_H_53_NO_2_ brown powder; 51.7% yield): IR (KBr, cm^−1^): 3405, 2933, 2850, 1718, 1461, 1267, 962, 721. ^1^H NMR (CDCl_3_) δH (ppm): 0.87 (m, 6H, CH_3_), 1.25 (m, 30H, CH_2_), 1.55 (m, 2H, CH_2_), 1.68 (m, 2H, CH_2_), 2.12 (t, 2H, CH_2_), 3.23 (s, 9H, CH_3_-N), 3.32 (t, 2H, CH_2_-N). ^13^C NMR (CDCl_3_) δH (ppm): 13.82 (CH_3_), 22.58 (CH_2_), 23.11 (CH_2_), 26.26 (CH_2_), 29.67 (CH_2_), 31.95 (CH_2_), 32.03 (CH_2_), 37.94 (CH_2_), 53.13 (CH_3_-N), 66.64 (CH_2_-N), 179.70 (C=O). 

Pentanoate 5-carboxy of N,N,N-trimethyl-hexadecan-1-ammonium ([THDA^+^][^−^AAD]; C_25_H_51_NO_4_; brown powder; 38% yield): IR (KBr, cm^−1^): 3425, 2919, 1691, 1467, 1282, 966, 719. ^1^H NMR (CDCl_3_) δH (ppm): 0.88 (t, 3H, CH_3_), 1.25 (m, 26H, CH_2_), 1.29 (m, 2H, CH_2_), 1.35 (m, 2H, CH_2_), 1.74 (m, 2H, CH_2_), 2.66 (s, 2, CH_2_), 2.85 (s, 2H, CH_2_), 3.02 (m, 2H, CH_2_-N), 3.37 (s, 9H, CH_3_-N). ^13^C NMR (CDCl_3_) δH (ppm): 14.07 (CH_3_), 22.64 (CH_2_), 23.14 (CH_2_), 24.21 (CH_2_), 29.08 (CH_2_), 29.67(CH_2_), 31,88(CH_2_), 32.85(CH_2_), 42.82 (CH_3_-N), 67.07 (CH_2_-N), 175.98 (C=O). 

Dodecanoate of N,N,N-trimethy-lhexadecan-1-ammonium ([THDA^+^][^−^AD]; brown powder; C_31_H_65_NO_2_; 45.8% yield): IR (KBr, cm^−1^): 3487, 2851, 1697, 1432, 1217, 910, 729. ^1^H NMR (CDCl_3_) δH (ppm): 0.88 (t, 6H, CH_3_), 1.25 (m, 44H, CH_2_), 1.61 (m, 2H, CH_2_), 2.32 (t, 2H, CH_2_), 2.87 (s, 2H, CH_2_), 3.37 (s, 9H, CH_3_-N). ^13^C NMR (CDCl_3_) δH (ppm): 14.04 (CH_3_), 22.63 (CH_2_), 23.13 (CH_2_), 24.80 (CH_2_), 27.93 (CH_2_), 29.57 (CH_2_), 31.86 (CH_2_), 34.17 (CH_2_), 42.97 (CH_3_-N), 66.94 (CH_2_-N), 178.23 (C=O).

Undecanoate 11-carboxy of N,N,N-trimethyl-hexadecan-1-ammonium ([THDA^+^][^−^A2D];C_31_H_63_NO_4_; brown powder; 42.7% yield): IR (KBr, cm^−1^): 3442, 2915, 2852, 1697, 1434, 1282, 927, 723. ^1^H NMR (CDCl_3_) δH (ppm): 0.86 (t, 3H, CH_3_), 1.23 (m, 40H, CH_2_), 1.55 (m, 4H, CH_2_), 2.21 (t, 4H, CH_2_), 3.24 (s, 9H, CH_3_-N), 3.36 (t, 2H, CH_2_-N). ^13^C NMR (CDCl_3_) δH (ppm): 14.09 (CH_3_), 22.62 (CH_2_), 25.63 (CH_2_), 29.28 (CH_2_), 31.85 (CH_2_), 36.03 (CH_2_), 53.10 (CH_3_-N), 66.68 (CH_2_-N), 178.27 (COOH, COO^−^).

3-Carboxybut-3-enoate of N,N,N-trihexyl-N-methyl-ammonium ([TXMA^+^][^−^AI]; C_24_H_47_NO_4_; brown viscous liquid; 96.1% yield): IR (KBr, cm^−1^): 3423, 3100, 2958, 2861, 1710, 1566, 1466, 1251, 942, 727, 556. ^1^H NMR (CDCl_3_) δH (ppm): 0.87 (t, 9H, CH_3_), 1.30 (m, 18H, CH_2_), 1.66 (m, 6H, CH_2_), 3.10 (s, 3H, CH_3_-N), 3.26 (m, 6H, CH_2_-N), 3.29 (s, 2H, CH_2_), 5.34 (s, 1H, C=CH_2_), 5.92 (s, 1H, C=CH_2_). ^13^C NMR (CDCl_3_) δH (ppm): 13.77 (CH_3_), 22.29 (CH_2_), 25.88 (CH_2_), 31.08 (CH_2_), 42.93 (CH_2_), 48.95 (CH_3_-N), 61.69 (CH_2_-N), 123.5 (H_2_C=C), 139.17 (C=CH_2_), 172.54 (COOH), 174.32 (COO^−^). 

3-Carboxy-2,2-dimethylpropanoate of N,N,N-trihexyl-N-methyl-ammonium ([TXMA^+^][^−^2,2-DSA]; C_25_H_51_NO_4_; brown viscous liquid; 96% yield): IR (KBr, cm^−1^): 3432, 2964, 2861, 1713, 1592, 1474, 1304, 1207, 977, 706. ^1^H NMR (CDCl_3_) δH (ppm): 0.87 (t, 9H, CH_3_), 1.20 (t, 6H, CH_3_), 1.30 (m, 18H, CH_2_), 1.66 (m, 6H, CH_2_), 2.47 (s, 2H, CH_2_), 3.12 (s, 3H, CH_3_-N), 3.30 (t, 6H, CH_2_-N), 3.64 (m, 1H, COOH). ^13^C NMR (CDCl_3_) δH (ppm): 13.76 (CH_3_), 22.29 (CH_2_), 25.89 (CH_2_), 26.65 (CH_3_), 31.08 (CH_2_), 40.91 (CH_2_), 48.47 (CH_3_-N), 61.65 (CH_2_-N), 176.26 (COOH), 181.93 (COO^−^). 

2-Amine-benzoate of N,N,N-trihexyl-N-methyl-ammonium ([TXMA^+^][^−^AA]; C_26_H_48_N_2_O_2_; brown viscous liquid; 95.7% yield): IR (KBr, cm^−1^): 3408, 2952, 2864, 1610, 1527, 1471, 1374, 1260, 1157, 753. ^1^H NMR (CDCl_3_) δH (ppm): 0.87 (t, 9H, CH_3_), 1.24 (m, 18H, CH_2_), 1.51 (m, 6H, CH_2_), 3.09 (s, 3H, CH_3_-N), 3.10 (t, 6H, CH_2_-N), 6.55 (m, 2H, Ar), 7.05 (t, 1H, Ar), 7.33 (s, 2H, Ar-NH_2_), 7.92 (d, 1H, Ar). ^13^C NMR (CDCl_3_) δH (ppm): 13.79 (CH_3_), 18.34 (CH_2_), 22.32 (CH_2_), 25.82 (CH_2_), 31.15 (CH_2_), 48.61 (CH_3_-N), 61.33 (CH_2_-N), 115.69 (C-Ar), 115.99 (C-Ar), 130.58 (C-Ar), 132.14 (C-Ar), 149.37 (Ar-NH_2_), 173.12 (COO^−^). 

3-Carboxybut-3-enoate of N,N,N-tripentyl-N-methyl-ammonium ([TPMA^+^][^−^AI**]**; C_21_H_41_NO_4_; brown viscous liquid; 81% yield): IR (KBr, cm^−1^): 3432, 2955, 2870, 1919, 1704, 1574, 1466, 1248, 942, 733, 562. ^1^H NMR (CDCl_3_) δH (ppm): 0.89 (t, 9H, CH_3_), 1.34 (m, 12H, CH_2_), 1.66 (m, 6H, CH_2_), 3.09 (m, 3H, CH_3_-N), 3.26 (m, 6H, CH_2_-N), 3.28 (s, 2H, CH_2_), 5.35 (s, 1H, C=CH_2_), 5.92 (s, 1H, C=CH_2_). ^13^C NMR (CDCl_3_) δH (ppm): 13.70 (CH_3_), 21.89 (CH_2_), 28.24 (CH_2_), 42.82 (CH_2_), 48.46 (CH_3_-N), 61.68 (CH_2_-N), 123.44 (C=CH_2_), 139.03 (C=CH_2_), 172.54 (COOH), 174.36 (COO^−^). 

3-Carboxy-2,2-dimethylpropanoate of N,N,N-tripentyl-N-methyl-ammonium ([TPMA^+^][^−^2,2-DSA]; C_22_H_45_NO_4_; brown viscous liquid; 85.6% yield): IR (KBr, cm^−1^): 3432, 3091, 2955, 2867, 1710, 1577, 1251, 942, 736, 544. ^1^H NMR (CDCl_3_) δH (ppm): 0.89 (t, 9H, CH_3_), 1.20 (s, 6H, CH_3_), 1.34 (m, 12H, CH_2_), 1.66 (m, 6H, CH_2_), 3.11 (s, 3H, CH_3_-N), 3.29 (t, 6H, CH_2_-N), 3.64 (m, 2H, CH_2_). ^13^C NMR (CDCl_3_) δH (ppm): 13.69 (CH_3_), 21.90 (CH_2_), 26.61 (CH_3_), 28.25 (CH_2_), 40.91 (CH_2_), 47.05 (CH_2_), 48.46 (CH_3_-N), 61.63 (CH_2_-N), 176.32 (COOH), 181.95 (COO^−^). 

2-Amine-benzoate of N,N,N-tripentyl-N-methyl-ammonium ([TPMA^+^][^−^AA]; C_23_H_42_N_2_O_2_; brown viscous liquid; 98% yield): IR (KBr, cm^−1^): 3405, 3064, 2952, 2873, 1919, 1616, 1533, 1371, 1262, 859, 753, 659. ^1^H NMR (CDCl_3_) δH (ppm): 0.85 (t, 9H, CH_3_), 1.18 (m, 6H, CH_2_), 1.26 (m, 6H, CH_2_), 1.48 (m, 6H, CH_2_), 2.98 (s, 3H, CH_3_-N), 3.08 (m, 6H, CH_2_-N), 6.54 (m, 2H, Ar), 7.02 (m, 1H, Ar), 7.33 (s, 2, NH_2_), 7.92 (m, 1H, Ar). ^13^C NMR (CDCl_3_) δH (ppm): 13.71 (CH_3_), 18.44 (CH_2_), 21.86 (CH_2_), 28.16 (CH_2_), 48.49 (CH_3_-N), 61.28 (CH_2_-N), 115.55 (C-Ar), 115.93 (C-Ar), 120.66 (C-Ar), 130.56 (C-Ar), 132.15 (C-Ar), 149.45 (Ar-NH_2_), 172.82 (COO^−^).

### 2.2. Rp- and Tafel-Polarization Analysis

Figure 1 shows the polarization-resistance (*Rp*) and corrosion-current-density (*i_corr_*) behavior of API X52 steel employing the ILs as CIs at 100 ppm in 0.5 M HCl by the Rp- and Tafel-polarization techniques, respectively. It can be observed that the presence of all the ILs in the corrosive medium increased the polarization resistance and decreased the corrosion-current density in the metal–electrolyte interface. The charge-transfer processes in the presence of ILs were affected by the formation of a protecting film that reduced the steel-mass loss, which is in good agreement with what has been reported on the evaluation of other ILs as CIs [13].

The group of ILs featuring the cation N,N,N-tripentyl-N-methyl-ammonium [TPMA^+^] reached the lowest *Rp* values of 522, 424, and 287 Ω cm^2^ for [^−^AI, ^−^2,2-DSA and ^−^AA], respectively, and accordingly, higher corrosion-current density than the rest of the compounds, thus evidencing less protection of the steel surface. These ILs possess pending groups in their chemical structure that are capable of forming coordinate bonds with the metal surface [33]; despite this fact, there was no synergistic effect between their ions favoring affinity toward the steel surface, which could be related mainly to the short aliphatic chain that promoted a higher degree of inhibitor-covered surface, which displaced water molecules. This fact reveals that the cation and anion chemical structures play a major role during the adsorption mechanism of CIs on the steel active sites in the corrosive medium. 

By comparing the group of ILs with cation [TPMA^+^] with respect to the cation [TXMA^+^], the importance of the molecular size of the IL cation becomes evident, for the difference between both cations is of one carbon atom in their three aliphatic chains, with five and six carbon atoms, respectively. However, this slight cation difference allowed for the following *Rp* (Ω cm^2^) relationships: 684 [TXMA^+^][^−^AI] > 522 [TPMA^+^][^−^AI], 646 [TXMA^+^][^−^2,2-DSA] > 424 [TPMA^+^][^−^2,2-DSA], and 794 [TXMA^+^][^−^AA] > 287 [TPMA^+^][^−^AA]. These results indicate that the increase in aliphatic-chain length favors the CI molecule orientation and adsorption on the steel active sites, giving higher *Rp* and lower current-density values, which implies a slowing down of the corrosion rate and a reduction in the resistive processes in the metal–solution interface. The interaction between [TPMA^+^] and [TXMA^+^] and the API X52 steel surface was limited by the molecular orientation of the hydrophilic part toward the steel surface. Some research works have confirmed that the length of the aliphatic chains in the inhibitor structure is an important factor for its *IE* [21,34].

Furthermore, in the ILs with cation [THDA^+^], it was observed that the number of carboxyl groups in the anions positively promoted the CI behavior of these compounds as follows: For anions with very long aliphatic chains, higher *Rp* values were obtained when there was just one carboxylic group, [^−^AD] > [^−^A2D]. However, this was not the case for anions with short aliphatic chains, where the presence of two carboxylic groups improved the *Rp* response, [^−^AI] > [^−^AB] and [^−^AAD] > [^−^AH]. The compound [THDA^+^][^−^AD] revealed the importance of the aliphatic chain (12 carbon atoms), whereas the IL [THDA^+^][^−^AI] confirmed that in short chains, the presence of two carboxyl groups, improves the performance of the ILs as CIs. According to the structures of the studied ILs, the *Rp* and *i_corr_* behavior is related to the distribution of the electron density in the molecule structure, mainly in the anions. 

Figure 2 shows the *Rp* and Tafel curves of API X52 steel in 0.5 M HCl at 100 ppm of some ILs; similar curves were obtained for the rest of the studied compounds. Regarding the ILs with cation [THDA^+^], they feature a significant difference with respect to the other ILs, for they had just a single 16-C-aliphatic chain; this change led to a decrease in the *Rp* slopes and the current density of the Tafel curves with respect to the cations [TXMA^+^] and [TPMA^+^], which led to a more efficient blocking of the active sites due to their adsorption on the steel surface, which promoted better properties as CIs [35]. From this group, the ILs with anions [^−^AI] and [^−^AA] presented outstanding results with the lower-current-density Tafel curves. However, the IL with the anion [^−^AA] could be adsorbed more easily on the steel surface due to the presence of heteroatoms such as nitrogen and oxygen (amine and carboxyl groups) and molecule double bonds, which could contribute with higher electron density, and with it, to a higher capacity to interact with the metal surface, thus forming more stable bonds [36]. The CI properties of these ILs with cation [THDA^+^] depend mainly on the anion chemical structure.

Table 2 presents the electrochemical parameters calculated by linear regression of Rp and linear extrapolation of the Tafel polarization curves of API X52 steel in 0.5 M HCl at 100 ppm of ILs: *Rp*, corrosion potential (*E_corr_*), anodic and cathodic Tafel slopes (*β*_a_ and *β*_c_), and *i_corr_*. The inhibition efficiency by *Rp* (*IE_Rp_)* and Tafel curves (*IE_Tafel_)* of the ILs are also reported in Table 2, which were calculated with Equations (1) and (2), respectively:(1)$$IE_{Rp}=\left(\frac{Rp^{CI}-Rp^{0}}{Rp^{CI}}\right)\times 100$$
(2)$$IE_{Tafel}=\left(\frac{i_{corr}^{0}-i_{corr}^{CI}}{i_{corr}^{0}}\right)\times 100$$
where the superindexes 0 and *CI* represent the absence and presence of inhibitor, respectively.

In Table 2, it can be observed that in the presence of ILs the *i_corr_* value was lower than that of the blank. Due to the presence of inhibitor, the redox reactions in the metal–corrosive-medium interface were affected by the blocking of the active sites by IL molecules on the steel surface. The displacement of the steel *E_corr_* in the presence of ILs with respect to the blank was of −8 mV toward the cathodic zone and of +15 mV toward the anodic zone. The *E_corr_* displacement range with inhibitor indicates that the inhibition process occurred through either a mass-transfer phenomenon in the metal–solution interface or geometrical blocking. In both cases, the active sites are occupied by IL molecules and the rate of the redox reactions is affected by the inhibition process [4,37]. This fact confirms the adsorption of the ILs on the surface of the API X52 steel in both active zones. For this reason, these new ILs can be classified as mixed-type CIs at the evaluated concentration. 

Table 2 shows that from the ILs with cation [THDA^+^], the compounds that displayed the lowest efficiencies were those featuring the anions [^−^MC], [^−^AB], and [^−^AH], which were below 54 ± 3%; the three anions had a short aliphatic-chain length and a single carboxyl group. As for the [THDA^+^] compounds with anions [^−^2,2-DSA], [^−^AAD], and [^−^A2D], the efficiencies were equal to 72, 72, and 68%, respectively; in comparison with the cation [TXMA^+^] and anion [^−^AA], whose efficiency was equal to 68%, the *IEs* were very similar, confirming that both ions affected the corrosion-inhibition properties of the ILs. The highest *IE* values of 79, 84, and 78% for [THDA^+^] with [^−^AI], [^−^AA], and [^−^AD], respectively, are related to the chemical structure of the cation and anion. These ion combinations in the ILs allowed their adsorption on surface-active sites and the formation of a physical barrier with nucleophilic and hydrophilic properties capable of repelling medium aggressive ions (O^−2^, OH^−^, and Cl^−^, among others) that promote the redox reactions of API X52 steel in 0.5 M HCl [38]. 

Table 3 shows the electrochemical parameters of the ILs with cation [THDA^+^] and anions [^−^AI], [^−^AA], and [^−^AD] at different concentrations. It can be observed that the *i_corr_* values diminished with the increasing concentration of the three evaluated ILs; this phenomenon is associated with higher availability of the CI molecules in the aqueous medium, affinity and orientation toward the metallic surface, and their interaction with corrosion-complex products. These phenomena control the steel redox reactions [39]. Furthermore, the *β_a_* and *β_c_* values at different CI concentrations do not present a well-defined trend, and some authors have related this behavior to mixed-type CIs [40]. Likewise, the *E_corr_* displacements of the ILs with respect to the blank toward more negative values confirm the behavior of the compounds as mixed-type CIs with cathodic preference [41].

Figure 3 shows the inhibition-efficiency (*IE)* behavior as a function of the concentration of the ILs with cation [THDA^+^] and anions [^−^AI], [^−^AA], and ^−^AD] as CIs of API X52 steel in 0.5 M HCl. It can be observed that the *IE* was directly proportional to the concentration; however, it is clear that at concentrations above 100 ppm, the *IE* fell slightly. Similar behavior patterns have been reported for ILs featuring carboxylic groups [33].

Even when the carboxylic groups have a rich electronic density and affinity for the metallic surface, an increase in the CI concentration (greater than 100 ppm) does not imply the growth of the EI, since the kinetics of the physicochemical phenomena between the IL chemical structure and steel surface is limited by the active sites of the metallic surface and CI geometric arrangement on this surface.

For the ILs with cation [THDA^+^] and anions [^−^AI], [^−^AA], and [^−^AD], the *IE* is a function of the length of the cation (hexadecyl) aliphatic chain and anion pending groups, which provide the nucleophilic and hydrophilic parts [42]. Although the [THDA^+^][^−^AD] compound has a longer linear chain, its *IE* was lower than those of [THDA^+^][^−^AI and ^−^AA], evidencing the importance of the anion chemical configuration during the steel corrosion-inhibition process. Finally, it can be concluded that two carboxyl groups in short-chain anions (2 amino-benzoate) and a very long alkyl chain improved the inhibition properties of the ILs with cation [THDA^+^].

### 2.3. Electrochemical Impedance Spectroscopy (EIS)

Figure 4a shows the Nyquist diagram obtained from the EIS tests for API X52 steel in 0.5 M HCl with and without [THDA^+^][^−^AA] at different concentrations. The presence of a semicircle can be observed, which suggests that the corrosion mechanism was controlled by charge transfer [43]. Likewise, all the systems presented capacitive loops with similar shapes, indicating that the addition of different inhibitor concentrations does not modify the adsorption mechanism [44]. Furthermore, an in increase in the semicircle size with the inhibitor concentration was evidenced, suggesting higher protection as a consequence of the growing resistance to charge transfer due to the formation of a film on the steel surface in acid medium [41,45].

With respect to the Bode diagram from the impedance module, it can be observed that at low frequencies, the impedance grew with the CI concentration, which indicates better protection against corrosion [46], whereas the phase angle increased in the presence of [THDA^+^][^−^AA] due to its adsorption on the steel surface, producing a surface covering in the metal–medium interface [37,47]. The presence of a single time constant, observed in the Bode diagram within a frequency interval ranging from 10^2^ to 10^3^ Hz, shows that the charge-transfer resistance is the phenomenon that prevails in the corrosion process in the presence of inhibitor [48]. 

Due to the fact that the Nyquist diagrams exhibited depressed semicircles, which are related to the roughness and heterogeneity of the metal surface, the experimental data were fitted by means of an equivalent electrical circuit with one constant phase element (Figure 5) [49]. The equivalent electrical circuit, known as a Randles circuit, presents one resistance to the solution (*R_s_*), one resistance to the charge transfer (*R_ct_*), and one constant phase element related to the electrical double layer (*CPE_dl_*) [48,49]. 

The results obtained from fitting the spectra to the circuit are reported in Table 4. It can be observed that the *R_s_* values displayed a slightly significant variation from 2 to 17 Ω cm^2^, which suggests that the ohmic fall of the experimental tests was minimal [37]. The *n* values indicate the homogeneity of the surface, where values close to 1 refer to a completely homogeneous surface [48,50]. In this study, *n* values from 0.86 a 0.9 were obtained, which suggests non-ideal capacitive behavior that is associated with the heterogeneity (irregularity) of the steel surface [7].

As observed in Table 4, *R_ct_* was directly proportional to the concentration, which reveals a diminution of the corrosion rate at high concentrations [51]. The highest *R_ct_* value was 1319 Ω cm^2^ at 100 ppm of [THDA^+^][^−^AA] with respect to that of the blank, which was 134 Ω cm^2^, indicating the formation of a protecting film on the steel surface due to the adsorption of IL capable of protecting the metal from the corrosive medium [52].

The inhibition efficiency by the EIS (*IE_EIS_*) technique was calculated by means of Equation (3) [53]: (3)$$IE_{EIS}=\left(\frac{R_{ct}-R_{ct}^{0}}{R_{ct}}\right)\times 100$$
where *R_ct_* and *R*^0^*_ct_* correspond to the charge-transfer resistance with and without CI, respectively. Table 4 shows a maximal inhibition percentage of 89.9% at 100 ppm of [THDA^+^][^−^AA].

### 2.4. Adsorption Isotherm

Different studies on the evaluation of CIs for the protection of alloys in corrosive media have reported that the CI molecules are adsorbed on the steel surface by means of a physical or chemical phenomenon [35]. In order to understand which kind of adsorption mechanism is involved, adsorption isotherm models are employed, where the CI surface-coverage degree (*θ*) is a function of the affinity that the IL molecules have with the steel surface through physical- and chemical-adsorption processes [13,54]. The *θ* values at different concentrations of the ILs [THDA^+^][^−^AI, ^−^AA, and ^−^AD] were calculated (*θ* = *IE*/100) from the data of the *Rp* and Tafel techniques and fitted with the Frumkin, Temkin, and Langmuir adsorption isotherm models [35,55]. The best fit was obtained with the Langmuir isotherm, expressed by Equation (4):(4)$$\frac{C}{\theta }=\frac{1}{K_{ads}}+C$$
where *C* is the IL concentration and *K_ads_* is the adsorption equilibrium constant. The *K_ads_* values were obtained by plotting *C/θ* vs. *C*, as shown in Figure 6, producing a good fit of the experimental data with a correlation coefficient (*R^2^*) close to unity. The obtained *K_ads_* values were 131.8, 179.9, and 34.6 M^−1^ for the ILs [THDA^+^][^−^AD, ^−^AA, and ^−^AI], respectively. These values confirm the spontaneous adsorption of the IL molecules on the metallic surface [56].

*K_ads_* is associated with the standard Gibbs free energy of adsorption (Δ*G*^0^*_ads_*), expressed in Equation (5), which is a thermodynamic parameter that is frequently employed to elicit the interaction type between a CI and metallic surface [57]:(5)$$\Delta G_{ads}^{0}=-RT\ln(55.5\times K_{ads})$$
where *R* is the universal gas constant and *T* is the absolute temperature (298.15 K). The Δ*G*^0^*_ads_* values obtained for the ILs [THDA^+^][^−^AD, ^−^AI, and ^−^AA] were −29.2, −26.5, and −29.9 kJ mol^−1^, respectively. The negative Δ*G*^0^*_ads_* values are related to spontaneous-adsorption processes between the IL molecules and metallic surface [1]. It has been widely reported that Δ*G*^0^_ads_ values between −40 kJ mol^−1^ and −20 kJ mol^−1^ imply a physicochemical-adsorption process. The anion in these ILs features a carboxylic group with rich electron density, and for this reason, they can work as active centers that make possible the adsorption on a steel surface [58]. In addition, their π electrons and free electrons of the oxygen atoms can form stable chemical bonds [27]. Notwithstanding, the additional combination of pending groups such as -NH_2_ and aromatic rings, like in the case of the IL [THDA^+^][^−^AA], can contribute to electron-density synergy and to the formation of π-type coordination bonds with the metallic surface [12]. Furthermore, thanks to a suitable orientation and position of the carboxyl group, a physical-adsorption process can occur through Van der Waals electrostatic-attraction forces [27,58].

### 2.5. Surface-Morphology Analysis

To confirm the protection of API X52 steel in 0.5 M HCl by the ILs evaluated as CIs, SEM-EDS surface analyses were carried out. The micrograph in Figure 7a corresponds to the steel surface in the absence of IL, where uniform corrosion damage and irregular topography can be observed; the O and Cl EDS signals were also higher than those in Figure 7b,c, which reveals the steel oxidation through medium aggressive ions (O^−2^, OH^−^ and Cl^−^) that provoke the formation of corrosion products such as oxyhydroxides, iron oxides, and iron chlorides. Figure 7b,c correspond to steel protected with 100 ppm of the ILs [THDA^+^][^−^AA and ^−^AD], respectively. In both micrographs, a regular topography with a slight presence of corrosion products can be observed. The protection of the metallic surface is evident due to the blocking of the active sites and displacement of water molecules by the adsorption of IL molecules on the steel surface, thus reducing the corrosion rate.

### 2.6. Computer-Simulation Analysis

The quantum chemical calculations of the ILs with the best *IE* helped better understand the adsorption mechanism through analysis of the reactive sites in each optimized structure, the energy of the molecular orbitals, and other quantum parameters. The inhibiting behavior of the ILs was studied by employing MEP charge distribution and the energy values of the highest occupied molecular orbital (*E_HOMO_*) and the lowest unoccupied molecular orbital (*E_LUMO_*). 

Table 5 shows the optimized structures and molecular-electrostatic-potential (MEP) map of the selected ILs. The structures of the IL anions presented characteristic double bonds (-COO^−^ and -COOH) with resonance, as obtained in other theoretical studies of similar structures based on different carboxylic acids, benzene rings, and/or combinations [59,60]. In the case of [^−^AA], it displayed a “flat” benzene ring due to the presence of two functional groups, [-COO^−^ and -NH_2_] [59]. For [^−^AI], the carboxylic groups yielded similar conformations to those described for some acids comparable to those employed during the synthesis [61]. As for the anion [^−^AD], the alkyl chain presented the linear-conformation characteristic of C > 10 chains [62,63]. In the case of cations, the alkyl chains showed the common conformation of ammonium groups that has been reported in other studies [63,64,65]. Regarding the interactions, the -COO^−^ bonds in the anions oriented themselves preferably toward the front of the cation methyl groups, as has been found for other anionic species interacting with alkylammonium structures [64,66,67].

As can be observed in Table 5, MEP analysis of the isosurface allowed the regions of the molecule that were reactive sites for electrophilic and nucleophilic attacks to be identified through a color scale [63,68]: The red regions are associated with the most electronegative atoms and resonant rings, which are the molecule sites that are more susceptible to electrophilic attacks, i.e., that cede *e*^−^ to the steel surface to form coordinate bonds with the empty Fe *d*-orbital. The blue zones are related to sites that are vulnerable before nucleophilic attacks, i.e., that accept e^-^ from other species located in the cation [N^+^], -NH_2_ of [^−^AA] and their adjacent carbon atoms. Finally, the green regions are characteristic of groups that are involved in neither nucleophilic nor electrophilic attacks because their configuration is saturated and are located mainly in surrounding H atoms, cation alkyl chains, and particularly in [^−^AD].

The data obtained from the MEP can be complemented by the analysis of molecular orbitals. Since HOMO is the external occupied orbital, it participates as an electron donor due to the presence of elements such as oxygen, which has a free electron pair and is, in general, located at the anion, which was the case of the three ILs analyzed in this work (Table 6). On the other hand, LUMO is the empty internal orbital that works as electron acceptor and is, in most cases, found in ammonium groups and the first adjacent carbon atoms [63]; however, in the particular case of [^−^AA], it was located in the same group because it presented negative and positive heteroatoms, which is in contrast with the evaluated ILs with carboxylic and amine groups.

Table 7 shows the quantum parameters obtained for the IL optimized structures. As for the molecule donor–acceptor activity, it was estimated by means of the energy difference between *HOMO* and *LUMO*, which is referred to as the energy gap (Δ*G_L-H_*): (6)$$\Delta G_{L-H}=E_{\mathrm{LUMO}}-E_{\mathrm{HOMO}}$$

According to the literature, low Δ*G_L-H_* values suggest higher donor–acceptor activity. Based on the values shown in Table 7, Δ*G_L-H_* displayed the following trend based on the anion: [^−^AA] < [^−^AI] < [^−^AD]. Such behavior is related to the number of heteroatoms and the anion complexity. The previous results obtained with respect to the Δ*G_L-H_* values are directly associated with the *IE*, for a low Δ*G_L-H_* value implies higher adsorption capacity of a molecule on a steel surface [63].

On the other hand, the highest values of the dipole moment (*μ*) displayed by the three ILs, in contrast with that of water (1.85 D), are associated with a higher tendency to replace water molecules adsorbed on a steel surface by means of dipole–dipole interactions [63,69].

### 2.7. Inhibition Mechanism 

An inhibition mechanism occurring between the ILs and the metallic surface was proposed to explain the protection action (Figure 8). It should be noted that steel-oxidation products, salts, and molecular hydrogen were formed at the anodic and cathodic centers during the normal corrosion process. The situation changed with the presence of ILs. Probably, the anodic centers of the steel surface interacted with the functional groups of the anionic part of the ILs, where specifically, the carboxylic groups formed either weak bonds or were exchanged with hydroxyl, chlorine, or oxygen ions present in the aqueous solution. At the same time, the cationic part succeeded in forming a precipitation complex with steel-oxidation products, helping diminish the active centers and mitigate the corrosion process. The characteristics of the IL ions that improve their properties for inhibiting metal corrosion are the following: (i) very long aliphatic chains in both anions and cations, (ii) the presence of two carboxyl groups in the anion, and (iii) heteroatoms such as nitrogen and oxygen present in the amine and carboxyl functional groups and double molecular bonds that increase the electron density in the ions, thus forming more stable bonds. Based on the points mentioned above, the synergistic effect exerted by ammonium cations with long aliphatic chains and carboxyl groups in the anionic part of the obtained ILs produced good inhibition results.

## 3. Materials and Methods

### 3.1. Synthesis of ILs

For the synthesis of the ILs, analytic-grade reagents with purity above 98% (Sigma-Aldrich, CDMX, Mexico) were used. The ILs were divided into 3 main groups based on the base tertiary amine: dimethyl hexadecyl amine, trihexyl-amine, and tripentyl-amine. The synthesis took place in two stages: (a) A trialkylammonium-methylcarbonate-derived IL was produced by the reaction between a tertiary amine and dimethyl carbonate with a 1:2 molar ratio using methanol as the reaction medium in a Parr ^®^ 4848 reactor at 160 °C for 6 h, and (b) the exchange of the methylcarbonate anion for any of the following carboxylic acids was carried out: 2,2-dimethylsuccinic acid (2,2-DSA), anthranilic acid (AA), hexanoic acid (AH), butyric acid (AB), itaconic acid (AI), dodecanoic acid (AD), dodecanedioic acid (A2D), or adipic acid (AAD). For this purpose, equimolar quantities of the methylcarbonate IL and corresponding acid were stirred using 40 mL of methanol as reaction medium at 40 °C for 30 min. Finally, methanol was removed under vacuum. The synthesis procedure was followed as in previous works [70].

### 3.2. Materials and Test Solutions

Analytic-grade hydrochloric acid and deionized water were employed to prepare the corrosive medium (0.5 M HCl). Initially, all the ILs were evaluated at a concentration of 100 ppm. Afterward, the compounds with the best *IE* were further evaluated at 25, 50, 75, and 200 ppm. The selected metallic material was API X52 steel. Prior to each electrochemical test, the metal surface was abraded with SiC emery paper No. 600-1200, followed by a cleaning and drying treatment [71]. In addition, the metal samples used in the SEM surface analyses were polished with 0.05 μm alumina to obtain a mirror-finish surface. The morphology of the surface with and without CI was analyzed using a JEOL-JSM-6300 microscope.

### 3.3. Electrochemical Techniques

A conventional glass cell was used in the electrochemical tests. This cell consisted of three electrodes: (a) a working electrode (API X52 steel) with a working area of 0.289 cm^2^, (b) a reference electrode (Ag/AgCl) placed within a Luggin capillary to reduce the ohmic drop, and (c) a counter electrode (high purity platinum, 99.9%). The performed electrochemical tests were *Rp* and Tafel, both from the open circuit potential (*E_OCP_*) obtained after 15 min. For *Rp* [72], the potential interval ranged from −25 mV to +25 mV vs. *E_OCP_*, and for Tafel [73], the interval ranged from −250 mV to +250 mV vs. E_OCP_, with both tests at a scanning rate of 0.166 mV/s. All the potentiodynamic tests were carried out in triplicate at 298 K (25 °C) in an aerated solution in a potentiostat/galvanostat model PGSTAT302N and employing the software NOVA 2.1.4. EIS tests were carried out at steady state at room temperature with a frequency range of 100 kHz to 10 mHz using a 5 mV sinusoidal signal once the *E_OCP_* was stabilized [74]. 

### 3.4. DFT Calculations

The inhibitory behavior of ILs was supported by quantum calculations. The three ILs with the best inhibitory behavior were structurally optimized by checking the optimal position without symmetry restriction and in the singlet state (M = 1). The computational calculations were developed under the density-functional theory (DFT) using the Gaussian 09W [75] software based on the B3LYP/6-311G level of the theory. Gauss view v6.0 was used for visualization and generation of input files. When obtaining the optimal structure, the molecular orbitals (MOs) and the dipole moment (*μ*) of each IL were analyzed.

## 4. Conclusions

The electrochemical results confirmed the importance of the chemical structure of the ILs and their role as mixed-type CIs. The obtained ∆G_ads_ values suggested a physicochemical-adsorption process on the steel active sites, which diminished the damage of the steel surface. 

The cation chemical configuration played a major role in the properties of the ILs evaluated as CIs, obtaining the following cation-based *IE* order relationship: [THDA^+^] > [TXMA^+^] > [TPMA^+^]. It became evident that the length of the aliphatic chain was a relevant factor in the inhibition of the corrosion of API X52 steel in 0.5 M HCl. 

The carboxyl group and the size of the anions were also important variables for good *IE*: two carboxyl groups in a short linear structure and just a single carboxyl group in longer linear structures improved the inhibiting properties of the ILs as CIs. 

The best *IE* results were displayed by the ILs [THDA^+^][^−^AA > ^−^AI > ^−^AD] because the anions, in addition to a carboxyl group, featured two amino-benzoate groups, which represented an additional carboxyl group and a very long linear alkyl chain. These features favored the interaction with the steel surface by forming chemical-type coordinate bonds.

## Figures and Tables

**Figure 1 ijms-24-07613-f001:**
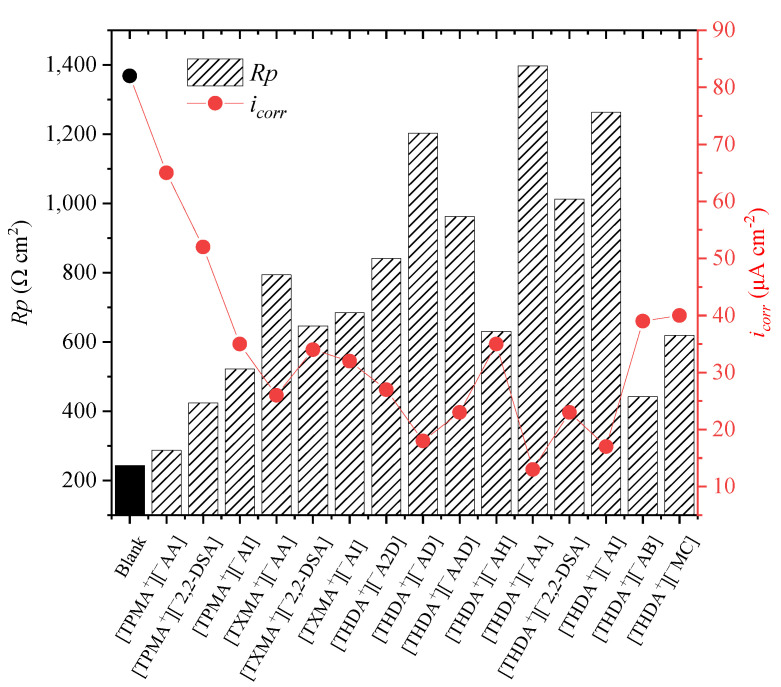
Rp and i_corr_ values of API X52 steel in 0.5 M HCl and ILs at 100 ppm by the Rp- and Tafel-polarization techniques, respectively.

**Figure 2 ijms-24-07613-f002:**
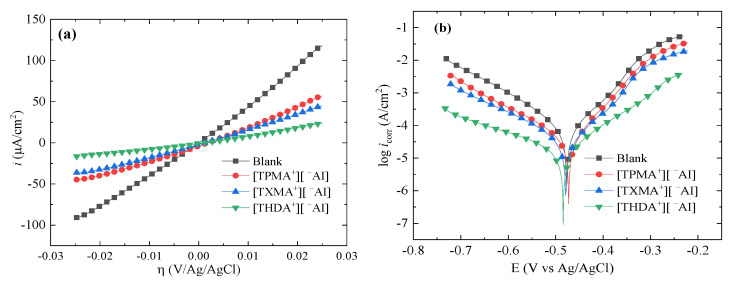
Rp and Tafel polarization of API X52 steel in 0.5 M HCl at 100 ppm of ILs with cations [TPMA^+^], [TXMA^+^], and [THDA^+^] and anions: (**a**,**b**) [^−^AI], (**c**,**d**) [^−^2,2-DSA], and (**e**,**f**) [^−^AA].

**Figure 3 ijms-24-07613-f003:**
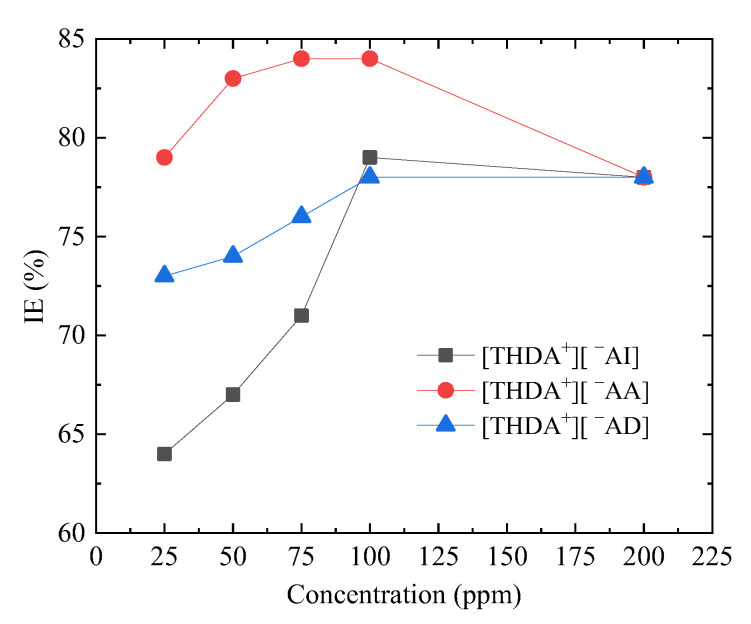
Corrosion IE behavior for API X52 steel in 0.5 M HCl at different concentrations of ILs with cation [THDA^+^].

**Figure 4 ijms-24-07613-f004:**
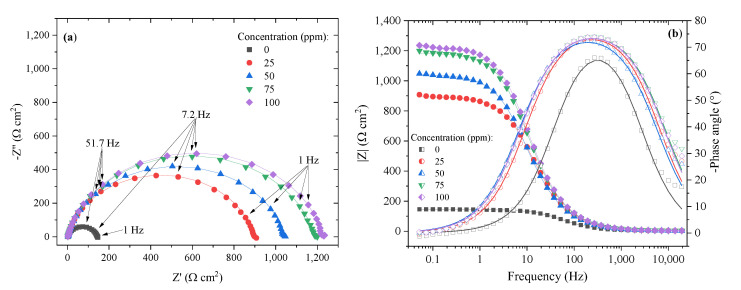
Impedance spectra of API X52 steel in 0.5 M HCl in the absence and presence of [THDA^+^][^−^AA] at different concentrations: (**a**) Nyquist and (**b**) Bode diagrams.

**Figure 5 ijms-24-07613-f005:**
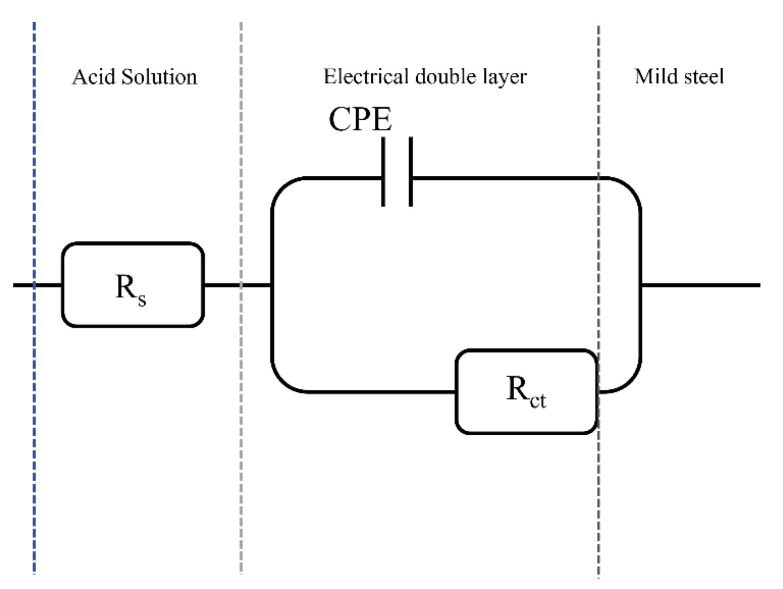
Equivalent electrical circuit of the API X52 steel system in corrosive medium in the presence of [THDA^+^][^−^AA].

**Figure 6 ijms-24-07613-f006:**
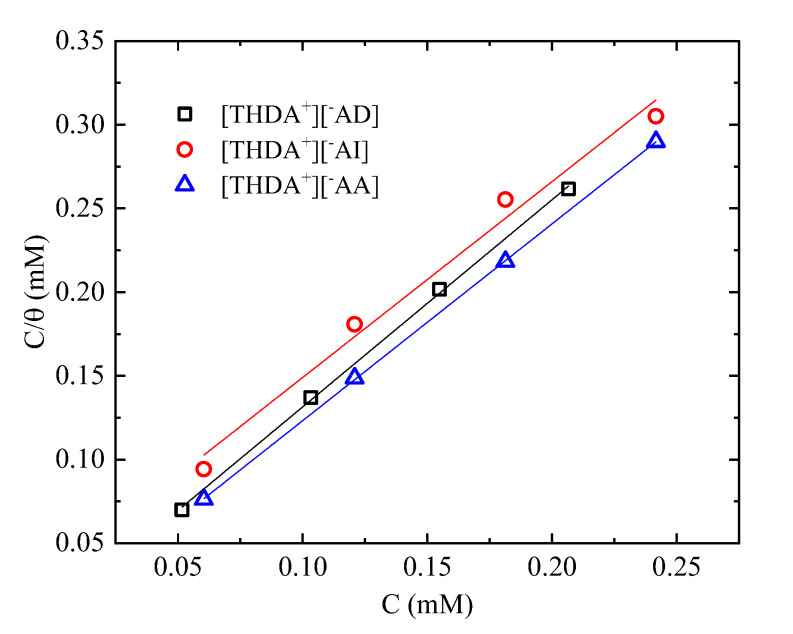
Langmuir adsorption isotherm for API X52 steel in 0.5 M HCl with ILs.

**Figure 7 ijms-24-07613-f007:**
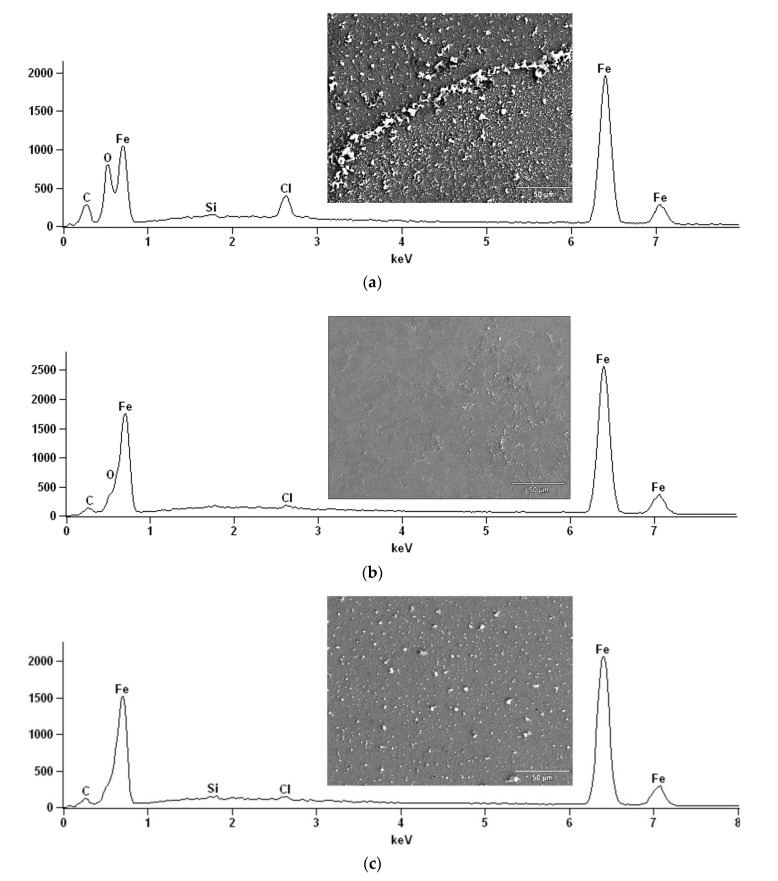
SEM-EDS micrographs of the API X52 steel surface after 2 h of immersion in 0.5 M HCl: (**a**) without IL and in the presence of (**b**) [THDA^+^][^−^AA] and (**c**) [THDA^+^][^−^AD].

**Figure 8 ijms-24-07613-f008:**
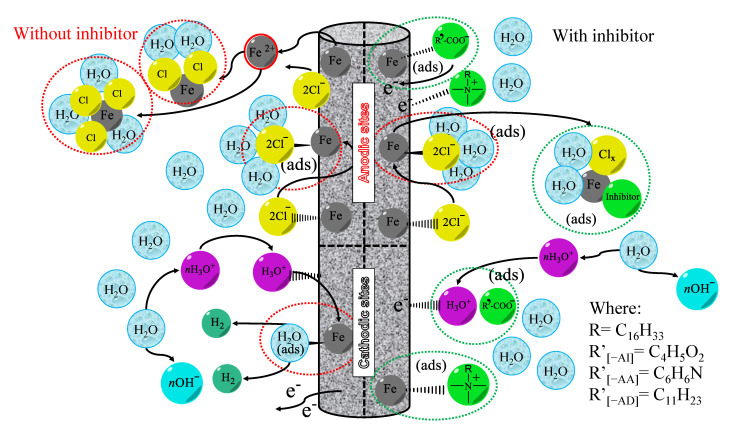
Inhibition mechanism carried out by the ILs evaluated as CIs of API X52 steel in 0.5 M HCl.

**Table 1 ijms-24-07613-t001:** Structures of the synthesized ILs.

Abbreviation	Name	Chemical Structure	Yield (%)
[THDA^+^][^−^MC]	Methyl-carbonate of N,N,N-trimethyl-hexadecan-1-ammonium	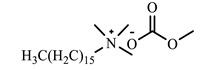	98.7
[THDA^+^][^−^AB]	Butyrate of N,N,N-trimethyl-hexadecan-1-ammonium	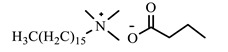	51.5
[THDA^+^][^−^AI]	3-Carboxybut-3-enoate of N,N,N-trimethyl-hexadecan-1-ammonium	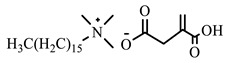	51.1
[THDA^+^][^−^2,2-DSA]	3-Carboxy-2,2-dimethylpropanoate of N,N,N-trimethyl-hexadecan-1-ammonium	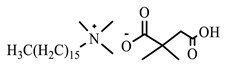	54.3
[THDA^+^][^−^AA]	2-Amine-benzoate of N,N,N-trimethyl-hexadecan-1-ammonium	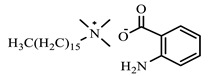	52.7
[THDA^+^][^−^AH]	Hexanoate of N,N,N-trimethy-l-hexadecan-1-ammonium	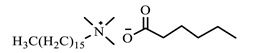	51.7
[THDA^+^][^−^AAD]	Pentanoate 5-carboxy of N,N,N-trimethyl-hexadecan-1-ammonium	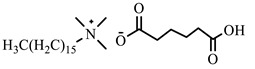	38.0
[THDA^+^][^−^AD]	Dodecanoate of N,N,N-trimethy-l-hexadecan-1-ammonium	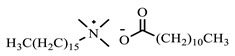	45.8
[THDA^+^][^−^A2D]	11-Carboxyundecanoate of N,N,N-trimethyl-hexadecan-1-ammonium	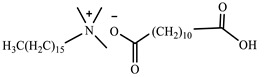	42.7
[TXMA^+^][^−^AI]	3-Carboxybut-3-enoate of N,N,N-trihexyl-N-methyl-ammonium	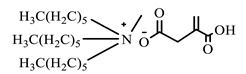	96.1
[TXMA^+^][^−^2,2-DSA]	3-Carboxy-2,2-dimethylpropanoate of N,N,N-trihexyl-N-methyl-ammonium	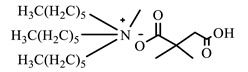	96.0
[TXMA^+^][^−^AA]	2-Amine-benzoate of N,N,N-trihexyl-N-methyl-ammonium	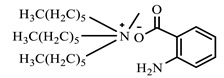	95.7
[TPMA^+^][^−^AI]	3-Carboxybut-3-enoate of N,N,N-tripentyl-N-methyl-ammonium	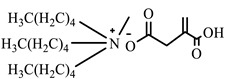	81.0
[TPMA^+^][^−^2,2-DSA]	3-Carboxy-2,2-dimethylpropanoate of N,N,N-tripentyl-N-methyl-ammonium	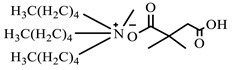	85.6
[TPMA^+^][^−^AA]	2-Amine-benzoate of N,N,N-tripentyl-N-methyl-ammonium	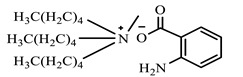	98.0

**Table 2 ijms-24-07613-t002:** Electrochemical parameters for API X52 steel in 0.5 M HCl at 100 ppm of ILs.

IL	*Rp* (Ω cm^2^)	*IE_Rp_* (%)	*E_corr_* (mV)	*β_a_* (mV dec^−1^)	*β_c_* (mV dec^−1^)	*i_corr_* (μA cm^−2^)	*IE_Tafel_* (%)
Blank	243 ± 9		−478 ± 1	89 ± 1	113 ± 2	82 ± 0.7	-
[THDA^+^][^−^MC]	619 ± 33	60.5 ± 4.0	−463 ± 7	106 ± 16	159 ± 13	40 ± 0.1	51.2 ± 0.4
[THDA^+^][^−^AB]	442 ± 0	45.1 ± 0	−478 ± 9	112 ± 9	144 ± 19	39 ± 0.4	52.4 ± 0.6
[THDA^+^][^−^AI]	1263 ± 26	80.9 ± 0.5	−482 ± 2	99 ± 6	214 ± 8	17 ± 0.8	79.2 ± 1
[THDA^+^][^−^2,2-DSA]	1013 ± 27	75.9 ± 1.9	−473 ± 2	98 ± 5	227 ± 3	23 ± 0.4	71.9 ± 0.5
[THDA^+^][^−^AA]	1397 ± 36	81.6 ± 2.3	−485 ± 2	81 ± 3	105 ± 8	13 ± 0.6	84.1 ± 0.7
[THDA^+^][^−^AH]	630 ± 31	61.4 ± 1.9	−476 ± 8	106 ± 15	137 ± 4	35 ± 0.4	57.3 ± 0.6
[THDA^+^][^−^AAD]	962 ± 32	74.7 ± 1.8	−486 ± 3	100 ± 1	223 ± 12	23 ± 0.6	71.9 ± 0.8
[THDA^+^][^−^AD]	1203 ± 37	79.8 ± 1.9	−481 ± 0	81 ± 5	231 ± 15	18 ± 0.2	78 ± 0.3
[THDA^+^][^−^A2D]	841 ± 29	71.0 ± 1.8	−474 ± 0	93 ± 2	224 ± 3	27 ± 0.1	67.1 ± 0.3
[TXMA^+^][^−^AI]	684 ± 39	63.9 ± 6.8	−480 ± 2	84 ± 8	110 ± 21	32 ± 0.5	60.9 ± 0.7
[TXMA^+^][^−^2,2-DSA]	646 ± 33	62.1 ± 4.0	−479 ± 4	88 ± 8	125 ± 7	34 ± 0.3	58.5 ± 0.5
[TXMA^+^][^−^AA]	794 ± 6	69.4 ± 0.2	−480 ± 3	83 ± 3	117 ± 10	26 ± 0.3	68.2 ± 0.5
[TPMA^+^][^−^AI]	522 ± 29	53.1 ± 3.3	−476 ± 6	75 ± 6	126 ± 2	35 ± 0.1	57.3 ± 0.4
[TPMA^+^][^−^2,2-DSA]	424 ± 15	38.6 ± 2.5	−477 ± 0	89 ± 16	125 ± 18	52 ± 0.8	36.5 ± 1.1
[TPMA^+^][^−^AA]	287 ± 0.7	15.3 ± 0.2	−477 ± 5	86 ± 12	134 ± 19	65 ± 1.4	20.7 ± 1.8

**Table 3 ijms-24-07613-t003:** Electrochemical data for [THDA^+^] and the anions [^−^AI], [^−^AA], and [^−^AD] at different concentrations.

IL	Concentration (ppm)	*E_corr_* (mV)	*β_a_* (mV dec^−1^)	*β_c_* (mV dec^−1^)	*i_corr_* (μA/cm^−2^)	*IE* (%)
Blank	0	−478 ± 1	89 ± 1	113 ± 2	82 ± 0.7	-
[THDA^+^][^−^AI]	25	−495 ± 1	130 ± 23	253 ± 6	29 ± 0.3	64.6 ± 0.5
	50	−499 ± 1	116 ± 3	240 ± 9	27 ± 0.1	67.1 ± 0.3
	75	−492 ± 1	102 ± 1	223 ± 3	24 ± 0.2	70.7 ± 0.3
	100	−482 ± 2	99 ± 6	214 ± 8	17 ± 0.8	79.2 ± 1
	200	−466 ± 7	78 ± 3	193 ± 6	18 ± 0.1	78.0 ± 0.2
[THDA^+^][^−^AA]	25	−491 ± 1	83 ± 1	98 ± 1	17 ± 0.2	79.3 ± 0.3
	50	−483 ± 5	80 ± 9	115 ± 8	14 ± 0.1	82.9 ± 0.2
	75	−483 ± 1	78 ± 2	111 ± 5	13 ± 0.3	84.1 ± 0.4
	100	−484 ± 2	81 ± 3	105 ± 8	13 ± 0.6	84.1 ± 0.7
	200	−466 ± 7	78 ± 3	112 ± 6	18 ± 0	78 ± 0.2
[THDA^+^][^−^AD]	25	−487 ± 1	93 ± 3	178 ± 18	22 ± 0.1	73.1 ± 0.3
	50	−489 ± 1	103 ± 3	187 ± 14	21 ± 0	74.4 ± 0.2
	75	−482 ± 0	89 ± 1	207 ± 19	19 ± 0.1	76.8 ± 0.2
	100	−481 ± 0	81 ± 5	231 ± 15	18 ± 0.2	78 ± 0.3
	200	−468 ± 5	76 ± 6	198 ± 8	18 ± 0.6	78 ± 0.8

**Table 4 ijms-24-07613-t004:** EIS parameters for the corrosion of API X52 steel in 0.5 M HCl in the absence and presence of [THDA^+^][^−^AA].

Concentration	*R_s_* (Ω cm^2^)	*Y* (μΩ^−1^ s^n^cm^−2^)	*n*	*R_ct_* (Ω cm^2^)	*IE_EIS_*
−	17 ± 0.2	53 ± 2.2	0.90	134 ± 3.4	−
25	3 ± 0	31 ± 5.1	0.88	889 ± 7.7	85.0 ± 1.0
50	3 ± 0.1	35 ± 2.7	0.86	1290 ± 15.3	89.7 ± 0.9
75	2 ± 0.1	32 ± 1.3	0.87	1149 ± 37.6	88.4 ± 1.2
100	3 ± 0.1	35 ± 11.9	0.86	1319 ± 88.6	89.9 ± 1.8

**Table 5 ijms-24-07613-t005:** Optimized structures and molecular-electrostatic-potential (MEP) mapping of the ILs obtained at B3LYP/6-311G level in aqueous medium.

IL	Optimized Structure	MEP
[THDA^+^][^−^AA]	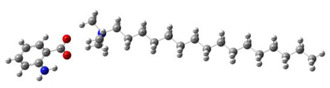	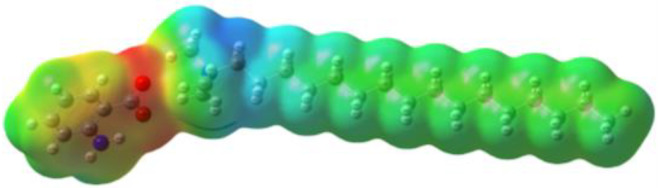
[THDA^+^][^−^AI]	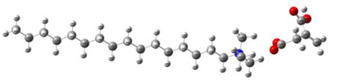	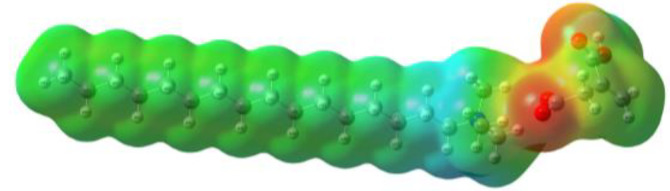
[THDA^+^][^−^AD]	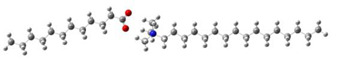	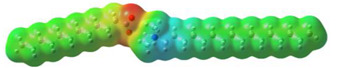

**Table 6 ijms-24-07613-t006:** Molecular orbitals of the ILs obtained at the B3LYP/6-311G level in aqueous medium.

IL	HOMO	LUMO
[THDA^+^][^−^AA]	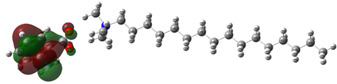	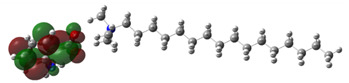
[THDA^+^][^−^AI]	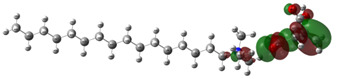	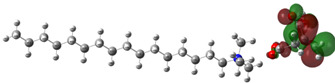
[THDA^+^][^−^AD]	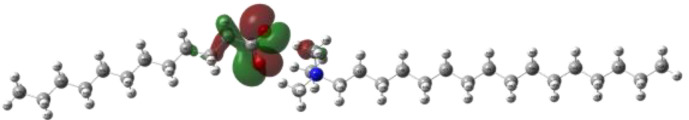	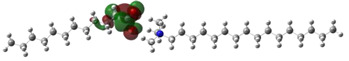

**Table 7 ijms-24-07613-t007:** IL quantum parameters at the B3LYP/6-311G level in aqueous medium.

IL	*E_HOMO_* (eV)	*E_LUMO_* (eV)	Δ*G_L-H_* (eV)	*μ* (D)
[THDA^+^][^−^AA]	−8.67	−5.37	3.31	15.39
[THDA^+^][^−^AI]	−9.96	−5.51	4.45	14.33
[THDA^+^][^−^AD]	−10.10	−3.63	6.46	14.37

## Data Availability

The data presented in this study are available on request from the corresponding author.

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
