# Peer review of "Synthesis of Ammonium-Based ILs with Different Lengths of Aliphatic Chains and Organic Halogen-Free Anions as Corrosion Inhibitors of API X52 Steel"

_ijms, 2023, doi:10.3390/ijms24087613_

Round 1

Reviewer 1 Report

The manuscript discusses the inhibition of the corrosion of API X52 steel in hydrochloric acid media by a series of ionic liquids at a ppm level. This reviewer did not find any major flaws in the manuscript. However, the Authors should address the following concerns:

Line 241 - what do the Authors mean by "the anion chemical configuration"?

Lines 293 - 295 - is this the short length that reduces the tendency of TPMA+ towards adsorption?

Line 297 - Are ionic liquids molecular or rather ionic compounds? What is adsorbed on the steel surface, IL molecules or perhaps ion pairs?  What is the evidence for IL anions not being protonated in HCl? What is the evidence for IL cations not forming ion pairs with Cl- in HCl? What is the evidence that there is no coadsorption of Cl- and IL cations or adsorption of protonated IL anions?

Lines 298 -300 - what is the evidence for the interaction between TPMA+ and TXMA+ being limited (?) or determined by the orientation of their hydrophilic parts towards the steel surface?

Line 307 - the acronym IE has to be explained, not everyone is a corrosion scientist

Lines 390 - 393 - the sentence is unclear

Line 425 - roughness and homogeneity or rather roughness and inhomogeneity?

Table 4 and Fig. 1 - why is the charge transfer resistance for 100 ppm of [THDA+][AA-] listed in the table very much different than the respective polarization resistance in Fig. 1?

Author Response

The authors thank the Reviewers for their precious time devoted to review the manuscript ijms-2327884. All the insightful observations and suggestions were taken into account to enhance the submitted paper according to the quality standards set by the Journal.

Reviewer #1

Line 241 - what do the Authors mean by “the anion chemical configuration”

Answer: Thank you very much for the observation. In order to avoid any confusion, the word “configuration” was replaced by “structure”, finally having: “the anion chemical structure”. To elaborate a little more on this point, in this research work, the cationic part (N,N,N-trimethyl-hexadecan-1-ammonium, THDA+) is the same for this series of ILs, and the only difference is the anionic part that is represented by 8 different chemical structures or chemical configurations. 3 [THDA +] ILs with anions such as dodecanoate [-AD], 2-amine-benzoate [-AA], 3-carboxybut-3-enoate and 3-carboxy-2,2-dimethylpropanoate [-AI] exhibited good properties as corrosion inhibitors (CIs), which could be attributed to the influence of the anion chemical structure along with the long aliphatic chain of the cationic part.

Lines 293 - 295 - is this the short length that reduces the tendency of TPMA+ towards adsorption?

Answer: The statement featured in lines 293-295 was modified in order to prevent any confusion.

Line 297 - Are ionic liquids molecular or rather ionic compounds? What is adsorbed on the steel surface, IL molecules or perhaps ion pairs?  What is the evidence for IL anions not being protonated in HCl? What is the evidence for IL cations not forming ion pairs with Cl- in HCl? What is the evidence that there is no coadsorption of Cl- and IL cations or adsorption of protonated IL anions?

Answer: The ammonium cation is not protonated, because nitrogen is positively charged and it does not have another available pair of electrons to form another bond. Since the ILs are utterly dissociated and solvated by water molecules in the aqueous solution, the formation of a new ionic pair is not likely because the hypothetical product of this reaction would also be soluble in water, being also dissociated and solvated. According to Langmuir isotherm, the CI is just adsorbed by a monolayer depending on the charge of the active site; the cations of the ILs are adsorbed on the cathodic sites, competing against hydronium cations, whereas the anions of the ILs are adsorbed on the anodic sites, competing with Cl- ions.

Lines 298 -300 - what is the evidence for the interaction between TPMA+ and TXMA+ being limited (?) or determined by the orientation of their hydrophilic parts towards the steel surface?

Answer: According to the molecular simulation study by C. Zuriaga et al., an increase in the length of aliphatic chains favors the surface approach. Then, it can be concluded that such an increase promotes hydrophobicity and the interaction between the surface and the IL. 

Zuriaga-Monroy, C., Oviedo-Roa, R., Montiel-Sánchez, L. E., Vega-Paz, A., Marin-Cruz, J., & Martinez-Magadan, J. M. (2016). Theoretical study of the aliphatic-chain length’s electronic effect on the corrosion inhibition activity of methylimidazole-based ionic liquids. Industrial & Engineering Chemistry Research, 55(12), 3506-3516.

Line 307 - the acronym IE has to be explained, not everyone is a corrosion scientist

Answer: We appreciate your observation. To be clearer, the acronym IE was defined in the manuscript.

Lines 390 - 393 - the sentence is unclear

Answer: The statement in lines 390-393 was modified in order to prevent any confusion.

Line 425 - roughness and homogeneity or rather roughness and inhomogeneity?:

Answer: This point was corrected as follows: “…roughness and heterogeneity of the metal surface…”

Table 4 and Fig. 1 - why is the charge transfer resistance for 100 ppm of [THDA+][AA-] listed in the table very much different than the respective polarization resistance in Fig. 1?

Answer: We do appreciate the Reviewer’s accurate observation. We reviewed in detail the results obtained by both techniques and realized that the area of the working electrode, in the EIS results, had not been introduced correctly, thus proceeding to the correction.

Reviewer 2 Report

The paper describe the performance of 15 ILs bases on ammonium cations and organic anions as corrosion inhibitors of steel in 0.5M HCl media. The authors make a wide electrochemical study and compare the corrosion resistance and efficiency obtained by means of polarization resistance, Tafel curves and EIS tests.

From my point of view, to be a totally sound corrosion study, there are several aspects of the work that should be reviewed:

1) Line 35: “before” it is not the correct adverb for this case. Also, steel pipes are not only stressed by shear forces.

2) Line 49: IL where not “developed” to be CI, they where first used as electrolytes.

3) Please, specify in the Introduction the novelty of these IL. Have they been used before?

4) Please, justify in the Introduction the selection of the 0.5M HCl media for this steel.

5) Figure 2 shows the sale information of Fig. 1, it is not necessary to duplicate this information as long as there is no new findings in Fig. 2

6) Line 265: The relationship of Rp and IL concentration is not evident at all in Fig 3. All curves are practically coincident.

7) As it can be expected, Rp and Tafel results are totally coincident, the comments of both results are totally identical and should be done together to avoid repetition. They are also some kind of confuse, and are disconnected of the final adsorption mechanism explanation. This aspect should be improved.

8) Table 4: What is the explanation for Rct value at 50 ppm? This high deviation is not acceptable and the test should be repeated and corrected.

9) Line 511: I don’t understand the reference in this sentence.

10) Please, complete SEM discussion with EDX analysis of the surfaces.

Author Response

+XCV

The authors thank the Reviewers for their precious time devoted to review the manuscript ijms-2327884. All the insightful observations and suggestions were taken into account to enhance the submitted paper according to the quality standards set by the Journal. Reviewer #2

1) Line 35: “before” it is not the correct adverb for this case. Also, steel pipes are not only stressed by shear forces.

Answer: We appreciate your observation. To be clearer, the word “before” was modified to “in case of” in the manuscript.

2) Line 49: IL where not “developed” to be CI, they where first used as electrolytes.

Answer: We appreciate your observation. To be clearer, the word “developed” was modified to “implementation” in the manuscript.

3) Please, specify in the Introduction the novelty of these IL. Have they been used before?

Answer: The Reviewer’s recommendation was taken into account and the work novelty was emphasized in the Introduction.

4) Please, justify in the Introduction the selection of the 0.5M HCl media for this steel.

Answer: The Reviewer’s suggestion was featured accordingly in the Introduction.

5) Figure 2 shows the sale information of Fig. 1, it is not necessary to duplicate this information as long as there is no new findings in Fig. 2

Answer: We merged the Rp and Tafel sections (Section 2.2) and then, Figs. 1 and 2 were modified accordingly. Furthermore, we consider that the featuring of Rp plots and Tafel curves is essential for better understand the results. 

6) Line 265: The relationship of Rp and IL concentration is not evident at all in Fig 3. All curves are practically coincident.

Answer: Despite the variation between the Rp and concentration values is minimal, a directly proportional relationship between these variables was observed (see figure below). Notwithstanding, based on the Reviewer’s suggestion, Fig. 3 was removed from the corrected manuscript in order to prevent information duplicity from happening.  

7) As it can be expected, Rp and Tafel results are totally coincident, the comments of both results are totally identical and should be done together to avoid repetition. They are also some kind of confuse, and are disconnected of the final adsorption mechanism explanation. This aspect should be improved.

Answer: The Reviewer’s suggestion was taken into account. To this end, a single section for the Rp results and Tafel curves is featured. In addition, the discussion regarding the electrochemical section and inhibition mechanism was clarified.

8) Table 4: What is the explanation for Rct value at 50 ppm? This high deviation is not acceptable and the test should be repeated and corrected.

Answer: An EIS test at 50 ppm was repeated and as a consequence, the deviation was reduced as shown in the following chart:

Concentration

Rs

(Ω cm2)

Y

(μΩ−1 sncm-2)

n

Rct

(Ω cm2)

IEEIS

17 ± 0.2

53 ± 2.2

0.9

134 ± 3.4

25

3 ± 0

31 ± 5.1

0.88

889 ± 7.7

85.0 ± 1.0

50

3 ± 0.1

35 ± 2.7

0.86

1290 ± 15.3

89.7 ± 0.9

75

2 ± 0.1

32 ± 1.3

0.87

1149 ± 37.6

88.4 ± 1.2

100

3 ± 0.1

35 ± 11.9

0.86

1319 ± 88.6

89.9 ± 1.8

9) Line 511: I don’t understand the reference in this sentence.

Answer: Reference (12) was verified to correspond to line 46 and it was eliminated from line 511 in the manuscript.

10) Please, complete SEM discussion with EDX analysis of the surfaces.

Answer: The EDS analysis was added to Section 2.5 “Surface morphology analysis” as suggested by the Reviewer.  

Reviewer 3 Report

In this manuscript, the authors synthesized different ionic liquids whit different structures (e.g., aliphatic chains and organic halogen-free anions) as corrosion inhibitors of API X52 steel in 0.5 M HCl. The corrosion inhibition of the tested inhibitors was analyzed by different electrochemical methods. A computational chemical approach was used to support the experimental results. Although the subject is interesting and the tested ionic liquids showed a good inhibition effect, there are some problems with this paper that should be addressed before it is considered for publication. They are outlined below:

1.      Lines 649-659. Why did you write this part in this section?  This part looks more like a conclusion, a conclusion that is completely missing in the manuscript.

2.      Section 2.2. Rather than reporting everything in terms of Rp, it is better to report the value of IE so that it can be compared with the other measurements.

3.      The temperature was not taken into account.

4.      No surface morphology results, except SEM-EDS, were presented. Product analysis should be conducted, including IFTR and/or XPS, etc.

Author Response

The authors thank the Reviewers for their precious time devoted to review the manuscript ijms-2327884. All the insightful observations and suggestions were taken into account to enhance the submitted paper according to the quality standards set by the Journal.

Reviewer #3

  1. Lines 649-659. Why did you write this part in this section? This part looks more like a conclusion, a conclusion that is completely missing in the manuscript.

Answer: We do thank the Reviewer’s accurate observation. To this end, this part was eliminated from Section 3 and added to the Conclusions.

  1. Section 2.2. Rather than reporting everything in terms of Rp, it is better to report the value of IE so that it can be compared with the other measurements.

Answer: The IE values from the Rp tests were added to Table 2 as suggested by the Reviewer.

  1. The temperature was not taken into account.

Answer: Dear Reviewer, the temperature was not considered as a variable for the present research work. We know that the redox processes increase with the temperature, however, many industrial processes take place at ambient temperature.

  1. No surface morphology results, except SEM-EDS, were presented. Product analysis should be conducted, including IFTR and/or XPS, etc.

Answer: We do thank the Reviewer’s observation, however, we would like to point out that at present we do not see the possibility of having immediate access to analysis equipment as suggested by the Reviewer. In addition, we consider that the employed electrochemical techniques and their corresponding analyses support satisfactorily our results and contribute to the study of the use of ionic liquids as corrosion inhibitors. 

Round 2

Reviewer 2 Report

The manuscript can be accepted in this version

Reviewer 3 Report

All questions have been satisfactorily answered. The manuscript may be published in its current form.